# Integrated Circuit of a Chua’s System Based on the Integral-Differential Nonlinear Resistance with Multi-Path Voltage-Controlled Oscillator

**DOI:** 10.3390/mi15030401

**Published:** 2024-03-16

**Authors:** Zhikui Duan, Huosheng Li, Shaobo He, Yongxi Long, Xinmei Yu, Qingqing Ke

**Affiliations:** 1School of Electronic Information Engineering, Foshan University, Foshan 528225, China; duanzhikui@fosu.edu.cn (Z.D.); 2112151039@stu.fosu.edu.cn (Y.L.); labxmyu@fosu.edu.cn (X.Y.); 2School of Automation and Electronic Information, Xiangtan University, Xiangtan 411105, China; 3School of Microelectronics Science and Technology, Sun Yat-Sen University, Guangzhou 510275, China; keqingq@mail.sysu.edu.cn

**Keywords:** Chua’s circuit, voltage-controlled oscillator, operational transconductance amplifier, chaos

## Abstract

In this paper, we present a fully integrated circuit without inductance implementing Chua’s chaotic system. The circuit described in this study utilizes the SMIC 180 nm CMOS process and incorporates a multi-path voltage-controlled oscillator (VCO). The integral-differential nonlinear resistance is utilized as a variable impedance component in the circuit, constructed using discrete devices from a microelectronics standpoint. Meanwhile, the utilization of a multi-path voltage-controlled oscillator ensures the provision of an adequate oscillation frequency and a stable waveform for the chaotic circuit. The analysis focuses on the intricate and dynamic behaviors exhibited by the chaotic microelectronic circuit. The experimental findings indicate that the oscillation frequency of the VCO can be adjusted within a range of 198 MHz to 320 MHz by manipulating the applied voltage from 0 V to 1.8 V. The circuit operates within a 1.8 V environment, and exhibits power consumption, gain–bandwidth product (GBW), area, and Lyapunov exponent values of 1.0782 mW, 4.43 GHz, 0.0165 mm^2^, and 0.6435∼1.0012, respectively. The aforementioned circuit design demonstrates the ability to generate chaotic behavior while also possessing the benefits of low power consumption, high frequency, and a compact size.

## 1. Introduction

Chaotic phenomena can be defined as the apparent lack of order and predictability in the motion of a deterministic system. A system, as described by deterministic theory, exhibits characteristics of uncertainty, irreproducibility, and unpredictability.

Three primary factors contribute to the occurrence of chaotic phenomena.

(1) Sensitive dependence on initial conditions, also known as chaotic attractors;

(2) Critical level, which refers to the threshold at which a nonlinear event takes place;

(3) Fractal dimension, which signifies the integration of order and disorder [1].

Since the 1980s, when Leon O. Chua introduced the first third-order autonomous chaotic oscillator and conducted research on systems owning double scrolls [2,3], the study of design, dynamics analysis, and implementation of chaotic systems have received much attention from the researchers in the fields of nonlinear science and cryptography [4,5,6,7,8,9,10].

Owing to its simple circuit structure and physical feasibility, the Chua circuit has been widely recognized as a prototype for studying chaotic phenomena and other nonlinear systems. The circuit is composed of an inductor, a resistor, two capacitors, and a nonlinear resistor. Until the present time, researchers have made substantial advancements in the investigation of Chua’s chaotic systems. Madan [11] introduced Chua’s circuit, a straightforward nonlinear electronic circuit design capable of demonstrating standard chaos theory behavior and exhibiting a multi-scroll chaotic phenomenon. Tang et al. [12] successfully generated a set of n-rolling attractors by employing a basic sine or cosine function, thereby demonstrating the occurrence of nine scrolls. Srisuchinwong et al. [13] employed a differential pair and a current mirror to fabricate a nonlinear resistor that exhibits characteristics akin to a Chua diode. Additionally, they integrated capacitance, inductance, and resistance to construct a Chua circuit. Meanwhile, the practical implementation of Chua’s system in real circuits plays a crucial role in its application, and various techniques are utilized for this purpose. In Ref. [14], a five-scroll chaotic circuit was designed, utilizing a segmented linear comparator constructed with Floating Gate MOSFET (FGMOSFET). Furthermore, the structure of the comparator, which was constructed using FGMOSFET, was enhanced. In the aforementioned article [15], a total of nine simple chaotic flows characterized by quadratic nonlinearity and the unique attribute of line equilibrium were identified. Tahir et al. [16] proposed the construction of a novel non-equilibrium chaotic system incorporating a state feedback controller capable of utilizing a multi-wing butterfly attractor. Multi-direction, multi-double-scroll Chua’s attractors can be generated by implementing pulsed excitation in all three dimensions of the relevant state variable [17]. Wang et al. [18] successfully produced multi-scroll attractors by incorporating memristors with a multi-piecewise continuous memductance function into Chua’s circuit. In the study conducted by Karthikeyan et al. [19], a parametrically controlled approach was employed to generate multi-scroll Chua’s attractors. Dai et al. [20] proposed a novel ternary fractal algorithm that expands the input sequences from two to three, resulting in a three-dimensional depiction of the chaotic attractor of a fractal transformation. The aforementioned study primarily examines the implementation of nonlinear resistors in a chaotic manner, including the utilization of multi-attractor chaotic implementation. However, limited emphasis is placed on the frequency generation aspect, with LC oscillators still being employed as the primary frequency generators. Zhang et al. [21] implemented a chaotic circuit using memristive discrete devices and FPGA to generate hyperchaotic phenomena. Valencia Ponce et al. [22] proposed a fractional-order chaotic-system-integrated circuit based on metaheuristic optimization. Wang et al. [23] proposed the implementation of two new five-element chaotic circuit structures using nonlinear circuits to control nonlinearity in 2023. In the same year, a numerical calculation method was used to study the hidden Cai’s attractor, and detailed system construction and equilibrium calculation were carried out, resulting in the visualization of the local attraction domain of a new multi-fold hidden Cai’s attractor [24]. The following year, a memristor was proposed to replace the Zea diode in the Zea circuit, enriching the circuit structure of the Zea circuit family [25]. Wu et al. [26] reported a new fractional-order Chua’s system with arctan function and an algorithm for determining the initial value of fractional-order hidden attractors. The DSP experimental results showed that the system can produce a pair of coexisting fractional-order hidden attractors, and the experimental results were consistent with the numerical simulation results. It verified the effectiveness of the presented methods and shows the potential engineering application value of the proposed Chua’s system. Zhao et al. [27] designed fully-fixed-point digital integrated circuits of discrete memristive systems with very low circuit area cost where the two-stage pipeline and multi-cycle architecture are employed. It provides the foundation for further applications of the discrete memristor and discrete memristor chaotic systems.

Chua’s circuit exhibits a straightforward circuit structure and is made physically feasible through the utilization of discrete components. In the domain of integration, the implementation process encounters difficulties primarily attributed to the LC oscillator [28]. Although the LC oscillator circuit employed in the circuit exhibits superior noise and frequency characteristics and is capable of achieving a high quality factor, it is accompanied by certain drawbacks such as occupying a significant amount of space, consuming high power, and exhibiting low output stability during design and application. These limitations contribute to increased chip design costs and reduced service life [29]. Eken et al. [30] provide a description of the design of voltage-controlled ring oscillators with three and nine stages capable of operating at frequencies up to 5.9 GHz. Increased switching speed and noise parameters have been observed. The voltage-controlled oscillator (VCO) discussed in the article [31] employs a CMOS current-mode logic stage to modify the dominant pole. This modification is performed to enhance the frequency response and enable the VCO to achieve higher oscillation frequencies when implementing a ring oscillator structure. Stadelmayer et al. [32] presented a comprehensive study on a broadband low-power synthesizer. The synthesizer is based on an eight-stage differential ring oscillator (RO) regulated by a Phase-Locked Loop (PLL). The authors successfully achieved a duty cycle close to 50% and ensured the same time delay for all RO stages.

In conclusion, it is possible to substitute the LC oscillator in Chua’s chaotic circuit with a VCO. This paper presents a novel approach utilizing CMOS technology to construct a multi-path VCO [33,34]. The proposed method aims to achieve a comparable performance to the LC oscillator circuit while also enabling the integration of discrete devices into the circuit [35]. The aforementioned solution addresses the limitations of the LC oscillator circuit while also enabling multi-segment output and expanding the tuning range within the realm of integration. The utilization of a VCO in lieu of an LC oscillator for the purpose of integrating Chua’s chaotic circuits not only results in a reduction in circuit size, but also broadens the range of applications, enhances circuit performance, and stimulates the advancement and investigation of integrated chaotic circuits. This approach effectively addresses the existing void in the field of integrated chaotic circuits.

The subsequent sections of this paper are structured in the following manner. Section 2 presents the construction of the conventional Chua circuit and the novel chaotic circuit’s construction without inductance and its mathematical model. Section 3 provides a comprehensive description of the design, principle, and all modules of the integrated chaotic circuit proposed in this research paper. Next, Section 4 presents the layout design and simulation results of the integrated chaotic circuit. Finally, the paper is concluded in Section 5.

## 2. The Proposed Chaotic Circuit

### 2.1. Structure of the Chua’s Chaotic Circuit

To demonstrate the presence of chaos in a self-excited circuit constructed using discrete components, it is necessary to satisfy three specific criteria [36]:

(1) One or more nonlinear components are required;

(2) One or more edge resistors are required;

(3) Three or more energy storage devices are required.

Chua’s circuit, which is referred to as “the paradigm of the chaotic circuit” [11], is the most basic chaotic circuit that fulfills the aforementioned conditions and demonstrates chaotic behavior. As depicted in Figure 1, Chua’s circuit incorporates an energy storage system comprising two capacitors, namely, $C_{1}$ and $C_{2}$, as well as an inductor *L*. These components are interconnected and coupled through an active resistor *R*. At the right side of the circuit, there is a nonlinear resistor denoted as $R_{N}$, also known as a Chua diode. This component is composed of three resistors and an operational transconductance amplifier (OTA).

### 2.2. Design of Chua’s Circuit without Inductance

We propose a new integrated chaotic circuit to implement Chua’s chaos without inductance and present the corresponding results in Figure 2. The new integrated chaotic circuit consists of a VCO and integral differential nonlinear resistors, which are coupled together through capacitance and resistance. The VCO is used to replace the original LC oscillator. The integral differential nonlinear resistance replaces the original Chua’s diode, and provides new variables to replace inductance variables.

The VCO presented in this study serves as a substitute for the inductance component in the conventional Chua circuit, yielding a comparable performance to that of the LC oscillator circuit. The proposed solution not only addresses the limitations of LC circuits, but also introduces a significant feature by enabling precise measurement of circuit delays and characterization of circuit aging. The VCO serves the purpose of creating signal delay, and it should be noted that the rate at which the capacitor $C_{2}$ receives the signal may vary, potentially resulting in the generation of oscillation. Meanwhile, the circuit contains two unstable points that interact with each other in order to achieve a balanced state, resulting in the production of a stable chaotic image. The introduction of $C_{1}$ results in the presence of an unstable point. Additionally, the nonlinear resistor $R_{N}$ delivers a continuous signal to the circuit. The negative impedance of $R_{N}$ induces oscillation in the circuit, and the inclusion of resistor *R* serves to interconnect the two circuit components, thereby creating a fully integrated dual-delay VCO chaotic circuit.

According to Figure 2, the state equation of the circuit can be expressed as follows:(1)$$\left\{\begin{array}{c} \dot{X}=\frac{1}{R_{N}C_{1}}Z+f\left(Z\right)\\ \dot{Y}=\frac{1}{R_{XY}C_{osc}}X+\left(\frac{1}{R_{XY}C_{osc}}+\frac{1}{R_{osc}C_{osc}}\right)Y\\ \dot{Z}=\frac{1}{R_{N}C_{1}}Z \end{array}\;\right..$$

The output voltage of the oscillator, denoted as X ($V_{Xout}$), and the voltage on the capacitor $C_{osc}$, which is used instead of $C_{2}$ in Chua’s circuit, are represented in the *X*-axis output. The voltage ($V_{Xout}$), connected in series between resistor $R_{XY}$ and capacitor $C_{0}$, serves as the *Y*-axis output and is used instead of R and $C_{1}$ in Chua’s circuit, denoted as Y ($V_{Yout}$). Then, the voltage ($V_{Yout}$), connected in series between $C_{0}$ and the nonlinear resistor, serves as the *Z*-axis output, denoted as Z. The function $f_{\left(Z\right)}$ is dependent on the nonlinear resistance, while the resistance $R_{osc}$ and $C_{osc}$ represent the internal resistance equivalent to the oscillator. The function $f_{\left(Z\right)}$ is defined as follows:(2)$$f_{\left(Z\right)}=\frac{R_{3}-R_{N}}{R_{0}R_{4}}Z+k\left(\frac{R_{3}-R_{N}}{R_{0}R_{4}}-\frac{1}{R_{0}}+\frac{1}{R_{3}+R_{N}}\right)\left(Z-|Z-E|\right)$$

The resistance network $R_{N}$ is composed of OTA2, $C_{1}$, $R_{1}$, and $R_{2}$, while *E* represents the saturated output voltage of OTA.

The primary obstacle faced by this design is the utilization of advanced integrated circuit technology to arrange these circuits on a solitary Integrated Chip (IC) in order to fulfill criteria such as a broad output frequency range and minimal power consumption. Compared to LC oscillator circuits, VCOs designed using complementary metal–oxide–semiconductor (CMOS) technology provide several advantages. These include a wide frequency range, lower power consumption, a compact layout area, and a stable output. The level of phase noise present in the output oscillation frequency is contingent upon the value of the VCO gain, denoted as $K_{VCO}$. The output frequency of the VCO can be adjusted by manipulating the delay time of the delay stage. The control delay time can be modified by manipulating the control voltage, denoted as $V_{C}$. In summary, the VCO is capable of adjusting its output frequency based on the applied voltage. It is then integrated into Chua’s circuit to generate chaotic phenomena.

In addition, the original LC oscillator was replaced by a VCO, and the state equation of the Chua’s system lacks the expression of inductance, requiring the introduction of new variables for replacement. Therefore, an integral-differential nonlinear resistance was proposed. By adding capacitors to the appropriate portion of the nonlinear resistor, the substitution of inductance for the Chua’s system can be achieved.

## 3. Further Explanation of the Basic Circuits

### 3.1. Proposal for Voltage-Controlled Oscillator in Integrated Chaotic Circuits

Figure 3 depicts the comprehensive circuit of the voltage-controlled oscillator (VCO) proposed in this research. The VCO comprises three five-stage loops and one seven-stage loop, which are constructed using inverters, variable capacitors, and XNOR as fundamental components. As depicted in the diagram, the seven-stage loop represents the slowest loop configuration, consisting of seven delay units. The system consists of one MOS tube and a five-stage loop, which is comprised of five delay units and one XNOR.

Figure 4 depicts the fundamental delay unit of the VCO proposed in this research. This unit is comprised of inverters and variable capacitors. Figure 5 depicts a XNOR circuit, which serves as the feedback mechanism for the VCO.

The phase noise in the output oscillation frequency of the multi-path VCO is contingent upon its gain, denoted as $K_{VCO}$. As the voltage-controlled oscillator (VCO) constant, denoted as $K_{VCO}$, decreases, there is an increase in the phase noise performance. The tuning range of the VCO will decrease as a consequence. To modulate the frequency of oscillation, the control voltage ($V_{c1}$) can be applied to the gates of the feedback PMOS in the seven-stage loop. The PMOS transistors serve as resistances, and the control voltage ($V_{c2}$) can be utilized to apply a variable capacitance. When the voltage, denoted as $V_{c1}$, increases, the resistance also increases, along with the parasitic capacitance. Consequently, the delay constant increases, leading to a decrease in the output oscillation frequency. Conversely, when the voltage decreases, the resistance, parasitic capacitance, delay constant, and output oscillation frequency exhibit the opposite behavior. Additionally, as the voltage $V_{c2}$ increases, the equivalent capacitance also increases, resulting in an increase in the delay constant and a decrease in the output oscillation frequency. Conversely, when the voltage decreases, the equivalent capacitance decreases, leading to a decrease in the delay constant and an increase in the output oscillation frequency.

The gain of the delay cells and the output frequency of the control voltage ($V_{c}$) can be mathematically represented as follows [37]:(3)$$f_{out}^{\prime }=f_{0}+K_{VCO}V_{C},$$
and
(4)$$K_{VCO}=\frac{f_{max}^{\prime }-f_{min}^{\prime }}{V_{max}-V_{min}}.$$

The symbol $f_{0}$ represents the output frequency. When the control voltage $V_{c}$ is equal to zero, the difference between the maximum and minimum values of $V_{c}$ ($V_{max}-V_{min}$) represents the range of variation. Additionally, *K* denotes a constant value. It is evident from the equation that a significant value of $K_{VCO}$ can be achieved, enabling precise control over the output frequency by manipulating a small variation in the voltage $V_{c}$. The characteristics of MOSFETs exhibit significant variations across different wafers. To mitigate challenges in circuit design, process engineers strive to ensure that device performance falls within a specified range. Roughly speaking, the expected parameter variation is tightly controlled within the maximum range that the chip’s performance envelope can tolerate. The range of performance typically provided to the designer is referred to as a “Process Corner”. Figure 6 illustrates the KVCO achieved through the simultaneous adjustment of VC1 and the fixation of VC2. As depicted in Figure 6, the three curves FF (blue), TT (red), and SS (yellow) correspond to Fast NMOS Fast PMOS, Typical NMOS Typical PMOS, and Slow NMOS Slow PMOS, respectively. In addition to the aforementioned three corners, it is also necessary for the three curves (FF, TT, SS) depicted in the figure to satisfy the requirements of voltage and temperature in order to establish PVT (process, voltage, temperature) conditions. The voltages are 1.8 V with a tolerance of +10%, 1.8 V, and 1.8 V with a tolerance of −10%, respectively. The recorded temperatures were −20 °C, 25 °C, and 80 °C, respectively.

Figure 7 illustrates the experimental results obtained from the output ports of the voltage-controlled oscillator (VCO) under the conditions where VDD and Vc are set to 1.8 V, where the output oscillation frequency is 32 MHz. Table 1 presents the dimensions of the CMOS that comprises the multi-path VCO element. The VCO circuit demonstrates several advantages, such as stability, high output frequency, and multi-stage output achievement.

### 3.2. The Nonlinear Resistor

#### 3.2.1. Circuit design of the Nonlinear Resistor

The nonlinear resistor is a device that exhibits non-Ohmic behavior when conducting electrical current. The deviation from a straight line in the volt–ampere characteristic curve indicates that the device no longer exhibits a linear relationship between voltage and current. However, it still adheres to Kirchhoff’s law. Generally, in the context of operational transconductance amplifiers (OTAs), it is common practice to connect the output of the OTA to its inverting input node in order to establish a negative feedback configuration. The utilization of negative feedback in a circuit is crucial for ensuring its stable operation. This is primarily due to the significant voltage gain exhibited by the operational amplifier.

As shown in Figure 8, it can be seen that there is an additional port for simple self-biasing of the reverse input of OTA2 compared to OTA1. This port connects to the Bias_out port in the OTA circuit in the final designed chip. In contrast, the inverting input of OTA1 is connected to the feedback signal and does not require a similar self-biasing port; hence, it is omitted in the circuit diagram. In fact, OTA2 can also adopt an external voltage for fixed biasing as long as it is adjusted to the appropriate voltage level to achieve the chaotic effect demonstrated in subsequent experimental results.

In this study, the design of a nonlinear resistance is achieved by utilizing two operational amplifiers, five resistors, and one capacitor, as depicted in Figure 8. Figure 9 and Figure 10 depict the current–voltage characteristic curves when the resistance or ratio of $R_{1}$ and $R_{2}$ is altered, respectively. The data demonstrate that the second current–voltage characteristic curve remains unaffected by the two parameters. The impact of varying total resistance values on the curve is evident in terms of curve stability, the rise and fall time of the first half peak, and the maximum value. The differential ratio influences the positioning of the first half of the current peak.

#### 3.2.2. Implementation of the Operational Amplifier

OTA refers to an integrated circuit that serves as a high-gain voltage amplifier specifically designed for differential-mode input, typically differential-in, single-ended output [38]. Its main function is to amplify the voltage difference between two input voltages [39]. The circuit is designed to operate efficiently under low-voltage conditions, ensuring a high output despite minimal input. The OTA proposed in this paper is depicted in Figure 11.

The input device generates the current, which is then amplified by a factor of B (B: the aspect ratio between the transistors) and ultimately passes through the output load. The output resistance, denoted as $R_{on}$, of the $V_{out}$ node is the sole high-resistance node within the circuit. Conversely, the resistances of the remaining nodes are approximately equal to $1/g_{m}$, where $g_{m}$ represents the transconductance of the transistor. This transconductance characterizes the transistor’s capability to represent voltage and convert current. Therefore, the present configuration can be classified as a two-stage amplifier, characterized by the presence of a single high-impedance node. Through the analysis, the output voltage of the circuit is expressed as:(5)$$V_{out}=\left(V_{p}-V_{n}\right)A_{v}.$$

Here, $A_{v}$ represents the gain of the stand-alone amplifier, and it is denoted as
(6)$$A_{V}=-G_{m}R_{on},$$
where $G_{m}$ represents the transconductance of the entire OTA, and $R_{on}$ is the output resistance. Meanwhile, $G_{m}$ can be expressed as
(7)$$G_{m}=2g_{m_{p}1}g_{m_{n}7},$$
and $R_{on}$ can be expressed as
(8)$$R_{on}=r_{o_{p}3}/\left[\left(1+g_{m_{p}7}B/g_{m_{n}5}\right)\left(g_{m_{n}1}+g_{m_{n}3}\right)\right].$$

In the aforementioned equations, the variables $g_{m}$, *r*, and *B* represent the transconductance, equivalent output resistance of the transistor, and the ratio of the width to the length of the transistors NM3 and NM5, respectively.

Table 2 presents the width-to-length ratio of the transistors in the OTA. According to the table, the width-to-length ratio of transistor PM0 is larger. This can be attributed to the fact that PM0 serves as the tail current source in the circuit, and the current passing through PM0 is the cumulative total of the differential circuit. Figure 12 presents the characteristic curve of the OTA amplitude–phase margin. The cascade OTA circuit exhibited a gain (20 lg (Av)) of 38.507 dB, a phase margin (r) of −131.28°, a 3 dB bandwidth of 221.65 MHz, and a gain–bandwidth product (GBW) of 4.23 GHz. The experimental results indicate that the power supply voltage (VDD) and the control voltage (Vb) were measured at 1.8 V and 1.2 V, respectively.

## 4. Chaotic Chip Layout and Its Simulations

### 4.1. The Chip Layout Diagram

This paper primarily focuses on the conversion of a chaotic circuit constructed using discrete devices into chips without inductance. The objective is to reduce power consumption and area while also enabling the realization of more intricate chaotic attractors. Experiments have demonstrated that the attainment of chaotic phenomena can be accomplished through the manipulation of device parameter configurations.

The following are the advantages of the proposed circuit.

Traditional Chua’s circuits are typically constructed using discrete components. The chaos circuit proposed in this article is completely integrated. The incorporation of this technology enhances the stability of the circuit while simultaneously minimizing the physical footprint required by conventional circuitry;The inductance L = 18 mH utilized in Chua’s circuit contributes to the generation of an oscillation frequency. The utilization of inductors results in higher power consumption; thus, the integration of chaotic circuits effectively addresses this issue;The Chua’s circuit constructed using discrete components is susceptible to external influences, leading to instability in the frequency of the inductor output. The VCO that has been proposed is completely integrated and demonstrates a high level of frequency stability;The integral differential nonlinear resistance circuit solves the problem of reducing system variables after replacing the LC oscillator with a VCO, and better realizes the complete integration of Cai’s chaotic circuit.

The chip layout area of the multi-path VCO chaotic circuit proposed in this article is 0.0165 mm^2^. The power supply voltage provided for the circuit is 1.8 V, and the corresponding power consumption is 1.0782 mW, as depicted in Figure 13. The frequencies of oscillation for the control voltage $V_{C}$ output range from 0.5 V to 1.8 V, with values of 320 MHz and 192 MHz, respectively. In this study, a fully integrated chaotic circuit is implemented using SMIC 180 nm CMOS tube technology.

### 4.2. Simulation Results

The measure of complexity in a chaotic system is determined by the proximity of the generated sequence to a random sequence. The anti-interference and anti-interception ability of the sequence is enhanced as its complexity increases. The comparison of complexity in chaotic systems is typically conducted through various methods, such as phase diagram observation and the measurement of Lyapunov exponents and entropy [40,41,42]. The observation of the phase diagram can only provide an approximation of the intricate nature of the chaotic system. By constructing the phase space and analyzing the resulting phase diagram, it can be observed that, as the complexity and chaos of the phase diagram increase, the overall complexity of the system also increases. The Lyapunov exponent quantifies the degree of sensitivity of a system to its initial conditions. If the exponent is greater than zero, it signifies that neighboring points will eventually diverge, leading to an unstable condition. The degree of chaos and system complexity increase as the exponent increases. A negative value signifies that neighboring points will ultimately converge to a single point, demonstrating the presence of a fixed point and periodic motion.

#### 4.2.1. Chaotic Phenomenon

To induce a chaotic phenomenon, the modulation of the width-to-length ratio of resistors, capacitors, and transistors in the circuit is indispensable. When the resistance and capacitance assume varying values, the system will exhibit diverse dynamic behaviors. In this study, the size of the transistor in the OTA remains constant, as depicted in Figure 2.

The experimental phenomenon of the integrated VCO chaotic circuit proposed in this paper is demonstrated in Figure 14 and Figure 15. The input voltage $V_{DD}$ is set to 1.8 V, and the bias voltage of OTA $V_{B}$ is set to 1.2 V.

Phase diagrams depicting various $X_{out}$−$Y_{out}$ relationships are presented in Figure 14, each corresponding to different parameter settings. Figure 14 illustrates the phase diagrams for three different scenarios. In the first scenario (a), the values of the resistors and capacitors are as follows: $R_{0}$ = 7 KΩ, $R_{1}$ = 21 KΩ, $R_{2}$ = 21 KΩ, $R_{3}$ = 1 KΩ, $R_{4}$ = 49 KΩ, $R_{XY}$ = 23 KΩ, $C_{0}$ = 50 fF, $C_{1}$ = 800 fF, and Ly = 0.8685. In the second scenario (b), the values are $R_{0}$ = 3 KΩ, $R_{1}$ = 10 KΩ, $R_{2}$ = 10 KΩ, $R_{3}$ = 7.1 KΩ, $R_{4}$ = 30 KΩ, $R_{XY}$ = 16.5 KΩ, $C_{0}$ = 200 fF, $C_{1}$ = 80 fF, and Ly = 0.6435. Finally, in the third scenario (c), the values are $R_{0}$ = 1 KΩ, $R_{1}$ = 10 KΩ, $R_{2}$ = 10 KΩ, $R_{3}$ = 2 KΩ, $R_{4}$ = 20 KΩ, $R_{XY}$ = 23 KΩ, $C_{0}$ = 100 fF, $C_{1}$ = 800 fF, and Ly = 1.0012.

Figure 14d depicts the current−voltage characteristics of the three nonlinear circuits under consideration. Comparing the phase diagrams and Lyapunov exponent, it is evident that, despite the apparent complexity of the phase diagrams, the Lyapunov exponent does not necessarily exhibit high values. This can be observed in Figure 14a,c. Therefore, it can be concluded that the requirement for achieving a chaotic response extends beyond the mere adjustment of coupling device parameters between the VCO and nonlinear resistors. Generally, the adjustment of parameters has the ability to alter the state of a system. This is analogous to modifying the values of various components in a circuit, including resistance, capacitance, and even bias voltage, in order to achieve desired outcomes. The observation of a phase diagram can provide an indication of the presence of chaotic behavior, but it cannot be considered as a conclusive method for determining the complexity of chaotic phenomena.

The phase diagrams of different $X_{out}$−$Y_{out}$ responses under experimental conditions are presented in Figure 15. In these experiments, the values of capacitance and the resistor were kept constant, while the frequency of the VCO was fixed. However, the values of $R_{3}$ and $R_{4}$ were varied. Comparing Figure 15b,c, it is evident that distinct resistance values have the potential to produce analogous chaotic phase diagrams. Figure 15c,d shows that only the resistance value of $R_{4}$ was modified, which varied from 20 kΩ to 30 kΩ. As a result, the chaotic phase diagram displayed consistent alterations.

#### 4.2.2. The Impact of Device Parameters on Chaotic Effects

In Figure 16, the phase diagram illustrates the transition in resistance value from below a certain range to above that range. No chaotic phenomena were observed in the phase diagram as the resistance values remained within the specified range, as shown in Figure 16a,d. The resistance between 20 K and 21 K leads to the emergence of chaos, which gradually intensifies, as depicted in Figure 16b. The phase diagram demonstrates evident chaotic phenomena when the resistance falls within the range of 21 k to 22.8 k, as depicted in Figure 16c. The chaotic phenomenon exhibits a gradual decline in intensity, as indicated by the resistance encountered between 22.8 k and 30 k, as depicted in Figure 16d,e. This observation demonstrates that, when the range above or below a certain threshold value, denoted as *R*, is reached, the system will exhibit either a stable or limited cycle state. Furthermore, it is evident that the variation of component values in a circuit has a consistent and predictable influence on the generation of chaotic phenomena.

To summarize, it can be inferred that determining an exact value as either large or small is not feasible. There is a specific range that exists. The range of component change within which the chaotic phenomenon is consistently affected. The intricate nature of the phase diagram does not necessarily imply the presence of significant chaotic effects. The verification of this requires the utilization of mathematical techniques, such as the Lyapunov exponent and bifurcation diagram, in order to combine and analyze the data.

### 4.3. Performance Comparison

Table 3 presents a comparative analysis of the current study and previous research conducted in the field of integrated chaos. It is evident from Table 3 that the study conducted by [35,43] presents chaotic circuits constructed using discrete devices. However, it is important to note that the performance of these circuits is not comparable to that of an integrated chaotic circuit in various aspects. Compared to other methods employed in the design of integrated chaotic circuits [29,30,39,44,45,46,47,48,49,50], the approach utilized in this study demonstrates superior performance in terms of circuit layout design and output frequency. It is indisputable that the integration of the circuit offers numerous benefits, and this study effectively showcases the advantages of the integrated chaotic circuit. Simultaneously, it can be observed that utilizing a smaller CMOS process yields improved experimental phenomena. This also contributes to the inevitable development trend of integration. Finally, it can be concluded that the integrated chaotic circuit surpasses the circuit constructed on the breadboard, aligning with the prevailing trend of development and meeting the market demands.

## 5. Conclusions

Currently, the utilization of integrated circuits is an unavoidable decision in numerous industries. In the conventional chaotic circuit, the utilization of discrete devices results in a bulky size and limited applicability. Additionally, the generation of chaos is significantly influenced by the surrounding environment. Due to the inherent sensitivity of chaotic circuits to initial conditions, there is a need for stringent requirements for discrete devices. To tackle the aforementioned concern, this research paper presents a novel solution to implement Chua’s chaos in the form of a chaotic integrated circuit. This circuit is designed using a multi-feedback loop-ring voltage-controlled oscillator. In addition, integral-differential nonlinear resistance is proposed for inductance-free implementation of Chua’s chaos. This phenomenon leads to a significant reduction in power consumption and circuit area, thereby expanding the potential applications and mitigating the influence of environmental factors on chaotic effects. Simultaneously, a comprehensive approach is employed to minimize the parameter error of the device and enhance the stability of the output chaotic signals. The circuit in this study is implemented using the 180 nm CMOS technology provided by the Semiconductor Manufacturing International Corporation (SMIC). The integrated circuit structure of this device enables it to achieve a wide output frequency range while maintaining a low power consumption of 1.0782 mW and occupying a chip area of 0.0165 mm^2^. Additionally, it exhibits high stability and provides multi-phase output. With the continuous reduction in CMOS feature size, there has been a growing trend in the development of and market demand for integrated circuits which has enabled the production of smaller and faster transistors. In the future, there will be increased interest and opportunities for chaotic chips in the realm of secure communication and other applications.

## Figures and Tables

**Figure 1 micromachines-15-00401-f001:**
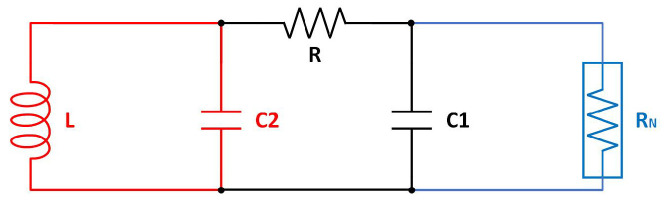
Typical Chua’s circuit.

**Figure 2 micromachines-15-00401-f002:**
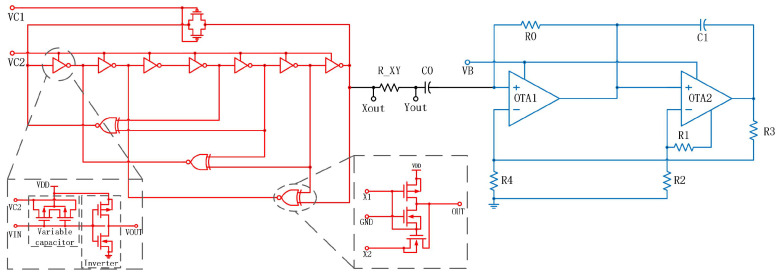
Fully integrated multi-path VCO chaotic circuit.

**Figure 3 micromachines-15-00401-f003:**
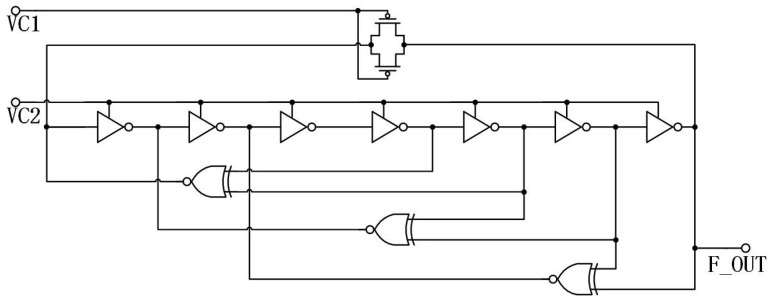
Multi-path voltage-controlled oscillator.

**Figure 4 micromachines-15-00401-f004:**
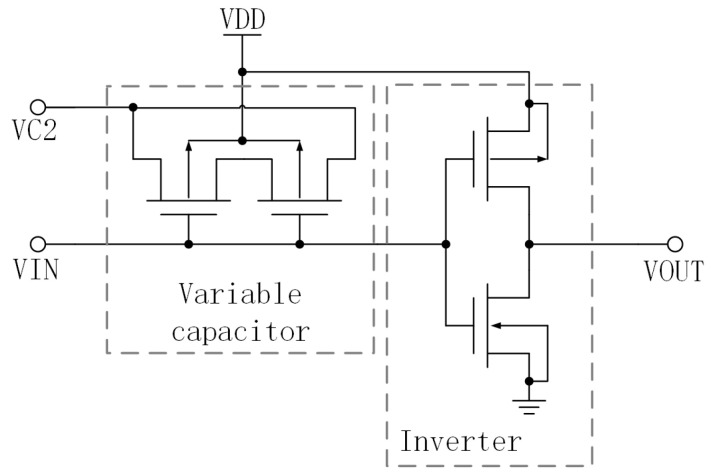
The basic delay unit.

**Figure 5 micromachines-15-00401-f005:**
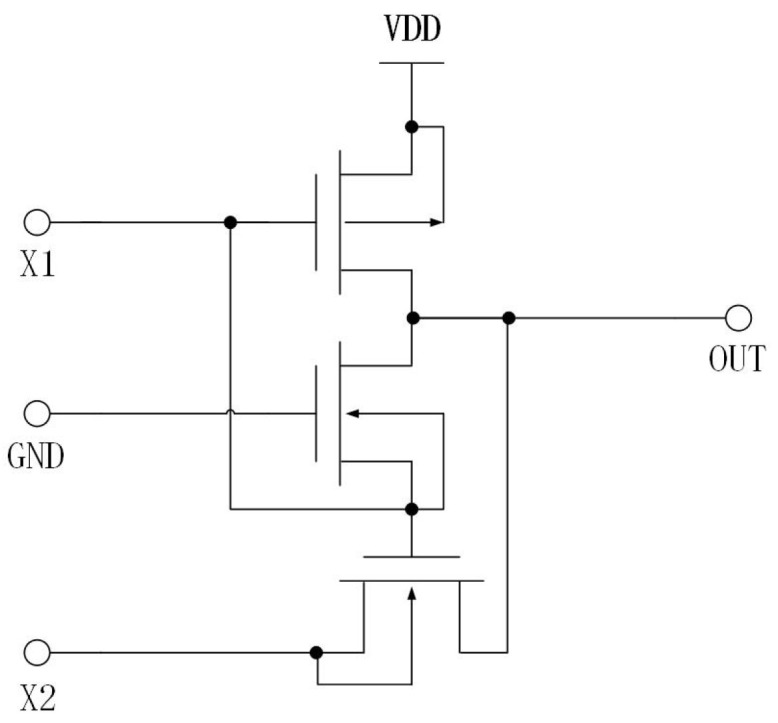
The feedback circuit.

**Figure 6 micromachines-15-00401-f006:**
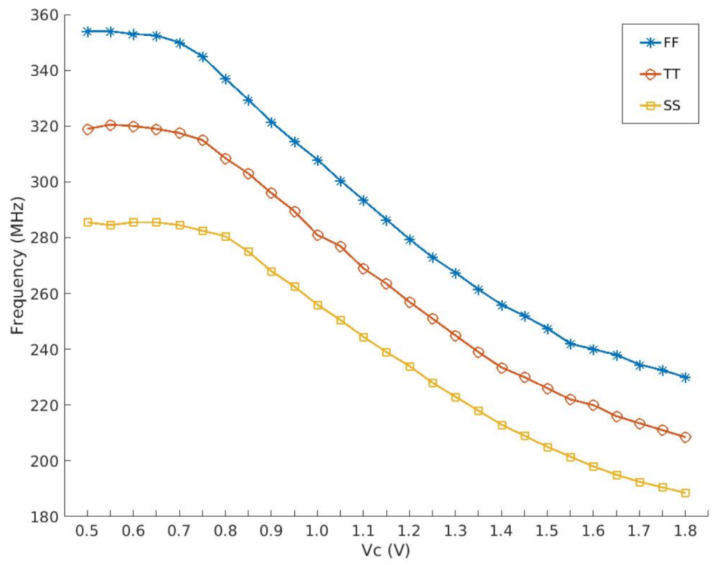
$K_{VCO}$ curves under different PVT processes.

**Figure 7 micromachines-15-00401-f007:**
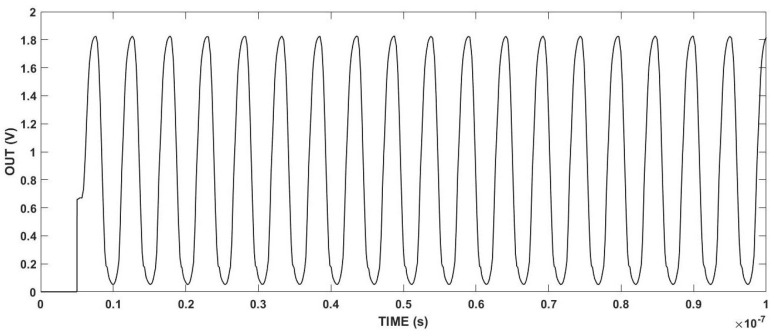
The output curve of voltage-controlled oscillators.

**Figure 8 micromachines-15-00401-f008:**
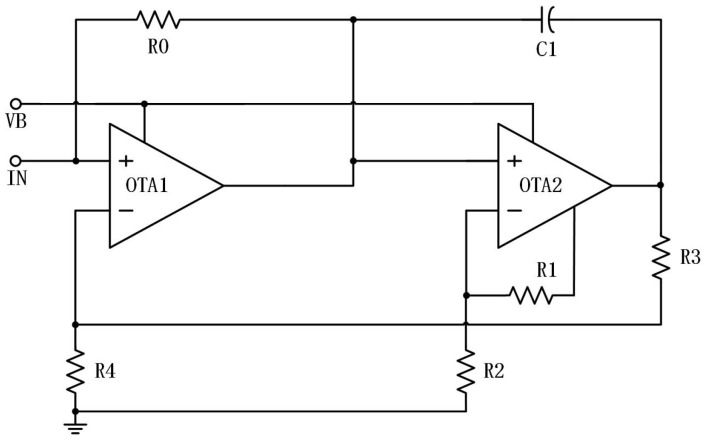
Nonlinear resistance.

**Figure 9 micromachines-15-00401-f009:**
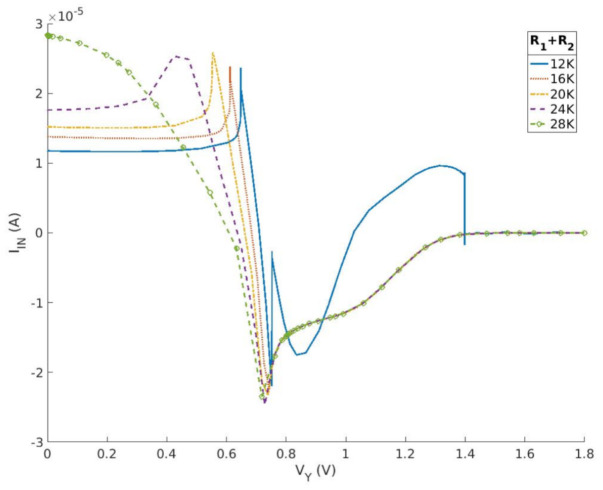
Current−voltage characteristics of proposed nonlinear circuit by differential resistance.

**Figure 10 micromachines-15-00401-f010:**
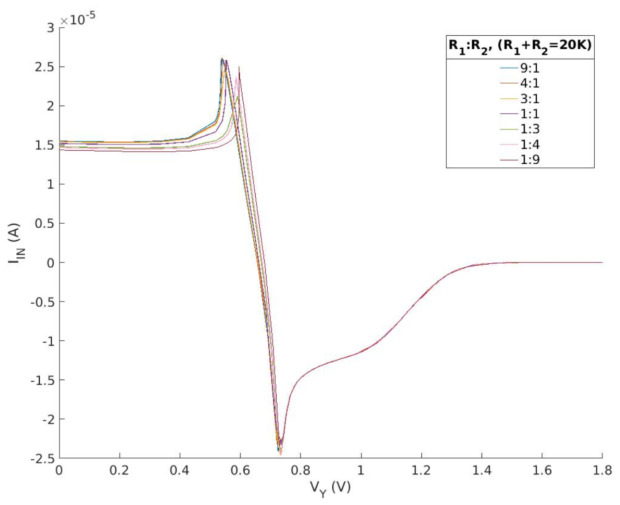
Current−voltage characteristics of proposed nonlinear circuit by differential resistance ratio.

**Figure 11 micromachines-15-00401-f011:**
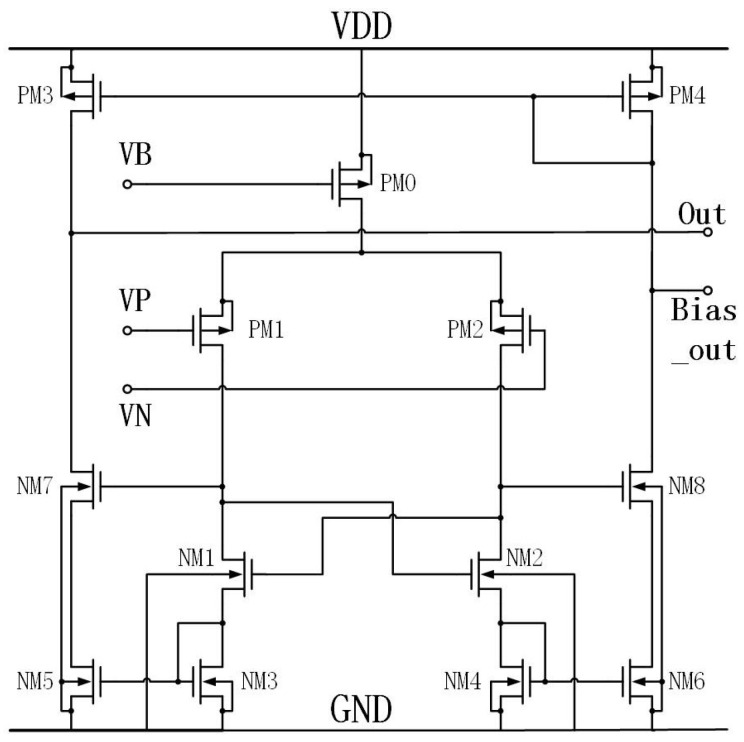
Operational transconductance amplifier.

**Figure 12 micromachines-15-00401-f012:**
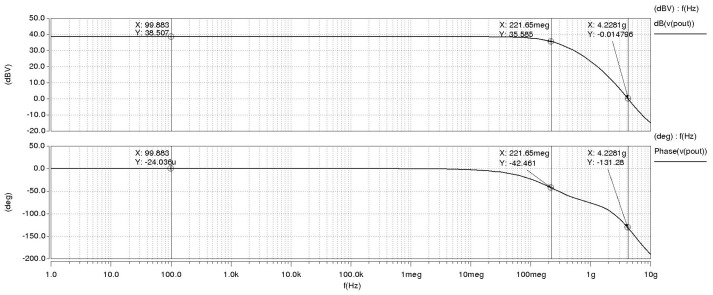
Bode plot.

**Figure 13 micromachines-15-00401-f013:**
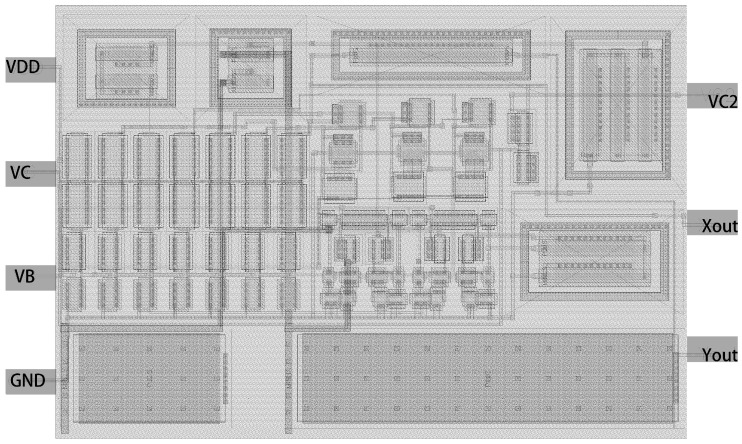
Integrated chaotic circuit layout.

**Figure 14 micromachines-15-00401-f014:**
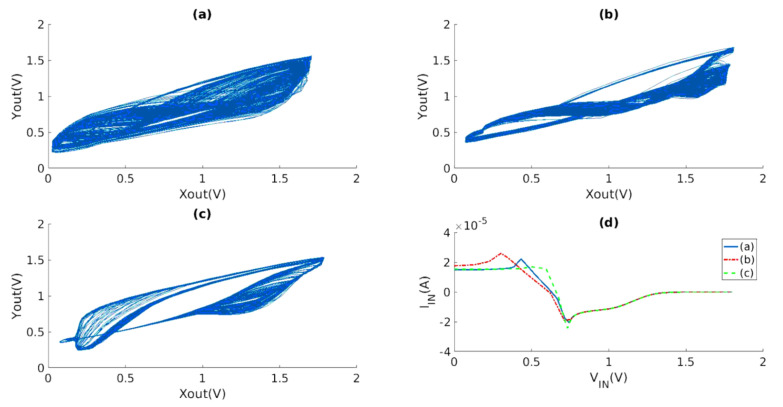
Chaotic phenomena by different nonlinear circuit parameters: (**a**) *f* = 220.62 MHz, $R_{0}$ = 7 KΩ, $R_{1}$ = 21 KΩ, $R_{2}$ = 21 KΩ, $R_{3}$ = 1 KΩ, $R_{4}$ = 49 KΩ, $R_{XY}$ = 23 KΩ, $C_{0}$ = 50 fF, $C_{1}$ = 800 fF, Lyapunov Index = 0.8685; (**b**) *f* = 196.89 MHz, $R_{0}$ = 3 KΩ, $R_{1}$ = 10 KΩ, $R_{2}$ = 10 KΩ, $R_{3}$ = 7.1 KΩ, $R_{4}$ = 30 KΩ, $R_{XY}$ = 16.5 KΩ, $C_{0}$ = 200 fF, $C_{1}$ = 80 fF, Lyapunov Index = 0.6435; (**c**) *f* = 303 MHz, $R_{0}$ = 1 KΩ, $R_{1}$ = 10 KΩ, $R_{2}$ = 10 KΩ, $R_{3}$ = 2 KΩ, $R_{4}$ = 20 KΩ, $R_{XY}$ = 23 KΩ, $C_{0}$ = 100 fF, $C_{1}$ = 800 fF, Lyapunov Index = 1.0012; (**d**) current–voltage characteristics.

**Figure 15 micromachines-15-00401-f015:**
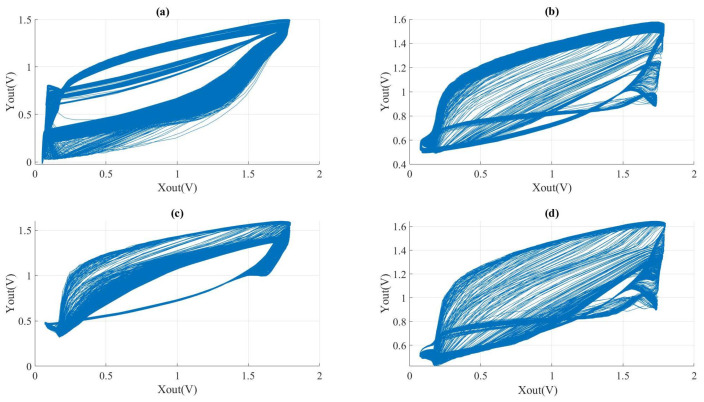
Chaotic phenomena by the same frequency (*f* = 303 Mhz) and different values of $R_{3}$ and $R_{4}$ ($R_{0}$ = 1 KΩ, $R_{1}$ = 10 KΩ, $R_{2}$ = 10 KΩ, $R_{XY}$ = 23 KΩ, $C_{0}$ = 800 fF, $C_{1}$ = 100 fF): (**a**) $R_{3}$ = 3 KΩ, $R_{4}$ = 39 KΩ, Lyapunov Index = 0.8976; (**b**) $R_{3}$ = 2 KΩ, $R_{4}$ = 20 KΩ, Lyapunov Index = 1.0012; (**c**) $R_{3}$ = 9 KΩ, $R_{4}$ = 25 KΩ, Lyapunov Index = 0.8221; (**d**) $R_{3}$ = 9 KΩ, $R_{4}$ = 22.5 KΩ, Lyapunov Index = 0.8539.

**Figure 16 micromachines-15-00401-f016:**
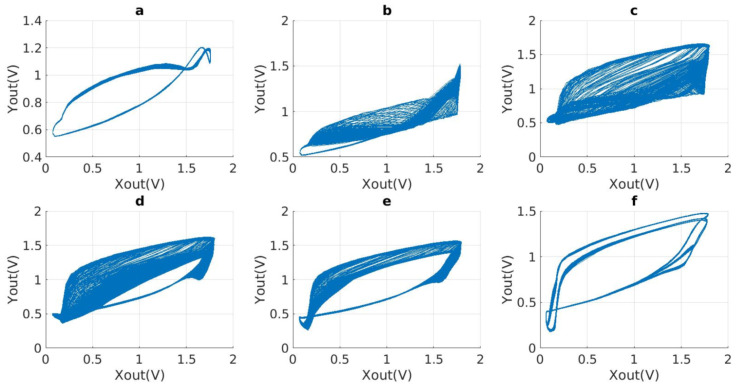
Chaotic phenomena by the same frequency (*f* = 303 Mhz) and different values of $R_{4}$ ($R_{0}$ = 1 KΩ, $R_{1}$ = 10 KΩ, $R_{2}$ = 10 KΩ, $R_{3}$ = 9 KΩ, $R_{XY}$ = 23 KΩ, $C_{0}$ = 800 fF, $C_{1}$ = 100 fF): (**a**) $R_{4}$ = 18.5 KΩ, Lyapunov Index < 0; (**b**) $R_{4}$ = 20.5 KΩ; (**c**) $R_{4}$ = 21 KΩ; (**d**) $R_{4}$ = 24 KΩ; (**e**) $R_{4}$ = 27 KΩ; (**f**) $R_{4}$ = 32 KΩ, Lyapunov Index < 0.

**Table 1 micromachines-15-00401-t001:** The size of the VCO’s basic elements.

Element	Transistor	Size (W/L (μm))	Transistor	Size (W/L (μm))
Delay Cell	NM1	1.0/0.2	NM2	1.0/0.2
	NM3	0.6/0.5	NM4	0.6/0.5
XNOR	NM5	1.0/0.2	NM6	1.0/0.2
	NM7	0.3/0.2	NM8	0.3/0.2
Transfer	PM0	3.0/0.2	PM1	1.5/0.2
	PM2	1.5/0.2	PM3	0.6/0.2

**Table 2 micromachines-15-00401-t002:** Transistor width–length ratio in cascade operational transconductance amplifier.

Transistor	Size (W/L (μm))	Transistor	Size (W/L (μm))
NM1	1.0/0.2	NM2	1.0/0.2
NM3	0.6/0.5	NM4	0.6/0.5
NM5	1.0/0.2	NM6	1.0/0.2
NM7	0.3/0.2	NM8	0.3/0.2
PM0	3.0/0.2	PM1	1.5/0.2
PM2	1.5/0.2	PM3	0.6/0.2
PM4	0.6/0.2		

**Table 3 micromachines-15-00401-t003:** The comparison of this work and former work.

[Ref. No.]	Architecture	Oscillation Frequency (MHz)	Supply Voltage (V)	Power (mW)	CMOS Process (μm)	Chip Area (mm^2^)
[21]	FPGA	6 (KHz)	-	-	Discrete	-
[30]	Single VDTA	20	±0.9	0.243	0.18	-
[22]	OTA	-	-	2.6	180 nm	-
[35]	FTICC	-	±15	-	Discrete	8.2 × 3.6 (cm^2^)
[39]	Chaotic PWM	1.2	3.3	-	0.18	0.626
[44]	True Random-Bit Generator	-	1.8	1.32	0.18	0.037
[45]	TRBG	100	1.8	0.9	0.18	-
[46]	Single-Delay VCO	0.011–0.036	1.8	2.0892	0.18	0.039
[47]	CFTA	9	±1.2	-	0.18	-
[48]	TS-CSK	-	1.8	1.5	0.18	1.5
[49]	Grounded Capactitors	-	0.65	0.252	0.13	0.12
[50]	IOT	6.25	2	4.5	0.18	2
This Work	Multi-Path VCO	320	1.8	1.0782	0.18	0.0165

## Data Availability

Dataset available on request from the authors.

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
