# Peer review of "Integrated Circuit of a Chua’s System Based on the Integral-Differential Nonlinear Resistance with Multi-Path Voltage-Controlled Oscillator"

_micromachines, 2024, doi:10.3390/mi15030401_

Round 1
Reviewer 1 Report
Comments and Suggestions for Authors
The paper describes the work clearly and accurately. The references are extensive and certainly adequate to support the presentation, especially at the introductory part (p.2). Yet I would advise a more careful review of the list as in some cases it seems that there is no connection to the subject of the paper (i.e which is the relevance of the content of ref 29 with this work??). Analysis of the proposed approach is also clear and correct. Section 3 describes almost all elementary things from the start till subsection 3.2.3, so this part (till 3.2.3) could be omitted as it is not at all useful in a state-of-the-art research publication. Other points of improvement: 1) line 54 in page 4 mentions authors and could easily misunderstood the refers to the author of the present paper, which is certainly not true, 2) in Fig 2 (and also in Fig 10) OTA2 is shown to have 5 pins for in/out/gm connections without any explanation instead of the usual for OTAs which is 4 pins (OTA1 has 4). Is this correct???? 3) Fig 14 presents a circuit diagram which is almost the same as appears in reference 39: Although ref 39 is mentioned at the start of the description, nowhere is mentioned the fact the circuit of Fig 13 is almost the same.
Author Response
Please find the response in the attached file. Thank you.

Reviewer 2 Report
Comments and Suggestions for Authors
This manuscript proposes a novel method and specific implementation for a Chua’s circuit using the integrated circuits without inductor components. Overall, this paper presents a novel integrated circuit design capable of generating complex chaotic behaviors. Moreover, this design offers advantages such as low power consumption, high frequency, and compact size, providing a new solution for the application of chaotic systems in the field of microelectronics. The present work is interesting and I think it can be accepted for publications in this journal. However, some minor revisions are listed as below.
1. Compared the original Chua’s circuit, the new circuit does not have inductor component but uses a multi-path voltage-controlled oscillator to drive the system. Are there any advantages in this new design?
2. To design the integrated chaotic circuits, there are different kinds of architectures been employed. It shows in table 3 that multi-path VCO employed in this manuscript has some advantages. Are there any other considerations for these new architectures?
3. The reference literature for performance comparison dates back quite old, such as the comparison of performance with discrete device-built chaotic circuits. Please check this issue.
4. Moreover, the literatures published in recent years are not sufficiant. The authors should add some more related work on Chua’s circuit and Chua’s system, such as dual Chua’s circuit, dual memristive Chua’s circuit, et al.
5. The paragraphs in the analysis of experimental structure are too lengthy and could be subdivided for clarity.
6. Please modify the English of this manuscript.
Comments on the Quality of English LanguageMinor editing of English language required
Author Response
Please find the response in the attached file.
